# Effect of dietary treatment and fluid intake on the prevention of recurrent calcium stones and changes in urine composition: A meta-analysis and systematic review

**Zhenghao Wang, Yu Zhang, Wuran Wei** * 

Department of Urology, Institute of Urology, West China Hospital, Sichuan University, Chengdu, China

* wzh2019kb@163.com

**Data Availability Statement:** All relevant data are within the manuscript and its Supporting information files.

**Funding:** The authors received no specific funds for this work.

## Abstract

To perform a systematic review and meta-analysis of randomized controlled trials (RCTs) for investigating the effect of dietary treatment and fluid intake on the prevention of recurrent calcium stones and changes in urine composition. PubMed, Web of Science, Embase, EBSCO, and Cochrane Library databases (updated November 2020) were searched for studies with the following keywords: diet, fluid, recurrent, prevention, randomized controlled trials, and nephrolithiasis. The search strategy and study selection process was conducted by following the PRISMA statement. Six RCTs were identified for satisfying the inclusion criteria and enrolled in this meta-analysis. Our result showed that low protein with or without high fiber diet intervention does not decrease the recurrence of stone upon comparing with control groups (RR = 2.32, 95% CI = 0.42–12.85; P = 0.34) with significant heterogeneity among the studies ($I^2$ = 81%, P = 0.02). But normal-calcium, low protein, low-salt diet had recurrences did reduced the recurrence compared to normal-calcium diet. And the fluid intake has a positive effect on prevention of recurrent stone formation (RR = 0.39, 95% CI = 0.19–0.80; P = 0.01) with insignificant heterogeneity among the studies ($I^2$ = 9%, P = 0.30). The different components of urine at baseline were reported in four studies. Upon reviewing the low protein with or without high fiber dietary therapy groups, it was found that there were no obvious changes in the 24-hour urine sodium, calcium, citrate, urea, and sulfate. In conclusion, our study shows that the only low protein with or without fiber does not affect recurrence, but low Na, normal Ca diet has a marked effect on reducing recurrence of calcium stone. And fluid intake shows a significant reduction in the recurrence of calcium stone.

## Introduction

Nephrolithiasis or urolithiasis is the third most common disease of the urinary tract that becomes more prevalent over the past decades [1, 2]. The worldwide kidney stone prevalence rate is 1.7% to 8.8% costs about $2.1 billion in 2020 [3, 4]. Patients with nephrolithiasis often suffer from short-term complications such as acute renal colic, nausea, vomiting, and

**Competing interests:** No authors have competing interests.

hematuria, and long-term complications such as chronic renal failure and hydronephrosis [5]. Furthermore, it is often associated with an incidence of recurrence after an initial event of 30% to 50% without prevention [6]. Therefore, to minimize the morbidity of nephrolithiasis, the reduction in calculi recurrence after the surgical clearance of stones is crucial.

The stones are mainly composed of 4 types of components—calcium oxalate, uric acid, calcium phosphate, and struvite, with calcium stones being the most common type [7]. The present study shows that the mechanism of stone formation is mainly based on metabolic defects which are combined from both genetic and nutritional factors [8]. Present efforts to prevent the recurrence of stones are mainly focus on changing the eurine compositions and decreasing concentrations of the lithogenic factors [9]. Currently, medication has a therapeutic effect on the prevention of recurrent stone formation by increasing renal calcium reabsorption, decreasing gut calcium absorption, chelating calcium in the urine or poisoning calcium crystal surfaces. Nevertheless, there are some associated side effects including gastrointestinal reaction, changes in blood pressure, and fluid/electrolyte imbalance [10]. Thus, some life-related prevention has been applied due to its high compliance and low side effects benefits [11]. To reduce calcium stone recurrence, preventive strategies targeting modifiable diet structure may be effective.

Previous studies have shown that formation of a renal stone is closely related to dietary regimes [12, 13]. Nevertheless, the higher evidence-based study like systematic review of clinical value in dietary therapy in urinary stone recurrence is still lacked. Furthermore, new studies with more detailed data at high evidence level are reported. Thus, we performed this systematic review and meta-analysis of randomized controlled trials (RCTs) for investigating the effects of dietary treatment and fluid intake on the prevention of recurrent calcium stones and changes in urine composition.

## Materials and methods

This systematic review and meta-analysis followed the guidelines of the Preferred Reporting Items for Systematic Reviews and Meta-analysis (PRISMA) statement and the Cochrane Handbook for Systematic Reviews of Interventions [14]. Ethical approval and patient consent were not required as all the analyses were based on previously published studies.

### Literature search and selection criteria

We systematically searched several databases including PubMed, Embase, Web of Science, EBSCO, and the Cochrane Library from the inception until November 2020 with the following keywords: diet, fluid, recurrent, prevention, randomized controlled trials, and nephrolithiasis. The reference list of retrieved studies and relevant reviews were hand-searched, and the process mentioned above was repeatedly performed for ensuring that all eligible studies were included. Inclusion criteria were as follows: (1) RCTs study design, (2) The patient had a stone history and was diagnosed by surgical removal, stone passage, or by imaging systems, (3) the intervention was diet or water intake, (4) adequate reporting of data provided for analysis, and (5) availability of the full text. All languages were included.

### Data extraction and outcome measures

Baseline information extracted from the original studies includes the first author, published year, number of patients, patient age and gender distributions, type of calcium stone, and detailed methods for the two groups, and the evaluation of evidence level. Data were independently extracted by two investigators and the discrepancies were resolved by consensus. The

primary outcomes were stone recurrence and withdrawal rate. The secondary outcomes were the variables of urine composition.

## Quality assessment of individual studies

All assessments were performed independently by two researchers with the differences resolved by the third researcher. the domain-based evaluation recommended by the Cochrane Handbook for Systematic Reviews of Intervention were used to address the following domains: bias arising from the randomisation process, bias due to deviations from intended interventions, bias due to missing outcome data, bias in measurement of the outcome and bias in selection of the reported result [15]. Figure of 'Risk of Bias' assessment were made by using Review Manager Software Version 5.3 (The Cochrane Collaboration, Software Update, Oxford, UK).

## Statistical analysis

Risk ratio (RR) with 95% confidence intervals (CIs) was calculated for dichotomous outcomes and heterogeneity was evaluated using the $I^2$ statistic, with $I^2 > 50\%$ indicating significant heterogeneity [16]. Sensitivity analysis was performed for evaluating the influence of a single study on the overall estimate by omitting one study or by performing subgroup analysis. The random-effects model was used for meta-analysis. Owing to the limited number of included studies (<10), publication bias was not assessed. Statistical significance was accepted at $P < 0.05$. All statistical analyses were performed using Review Manager Software Version 5.3 (The Cochrane Collaboration, Software Update, Oxford, UK).

## Results

### Literature search, study characteristics, and quality assessment

A total of 71 articles were initially identified from the database search. After the removal of duplicates, 45 articles were retained. Of these, 34 were excluded from analysis following the screening of the abstracts and titles, three were excluded as they were review articles, one was excluded because of insufficient data, and one was excluded because of the unavailability of the full text. Finally, six RCTs were identified as those satisfying the inclusion criteria and were finally enrolled in this meta-analysis [17–22]. The article selection process was performed by following the PRISMA guidelines (Fig 1). Baseline characteristics of the six included RCTs are shown in Table 1. These studies were published between 1996 and 2008, and the total sample size was 824.

Interventions of four studies take the dietary method [17, 19–21]. The study by Dussol *et al.* has two subgroups: low animal protein and high fiber. Patients in the study by Borghi *et al.* received a low protein and low salt diet [19]. Kocvara *et al.* take a special regime based on a metabolic evaluation which involves low protein, purine, oxalate, and high fiber [20]. Hiatt et al. compared the low protein and purine with fluid intake only [21]. The other two studies take water intake to make the urine volume higher than 2 L and 2.5 L, respectively [18, 22].

### Quality assessment of included studies

All studies had low risk bias in incomplete outcome data, selective reporting and other issues. Three studies [18, 20, 22] had high risk in blindness and random sequence generation while the other three studies [17, 19, 21] were not which is shown Fig 2.

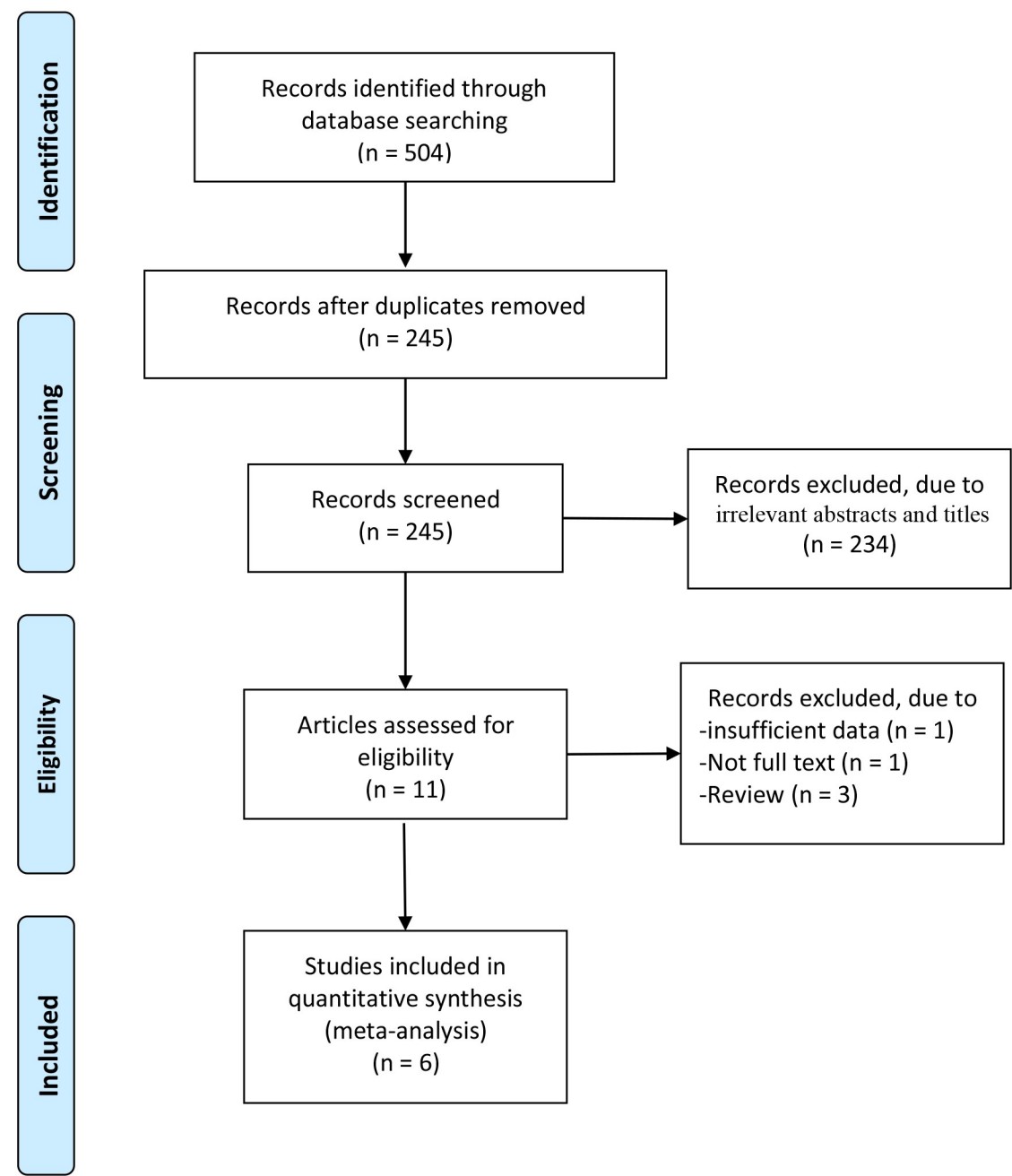

**Fig 1. Flow diagram of study searching and selection process.**

## Primary outcome: Recurrence rate and withdrawal rate

Totally Four studies reported the recurrence of the stones by the dietary method [17, 19–21] and two by the fluid method [18, 22]. Due to interventions for control group of Kocvara et al. [20] are tailored. Therefore, their result of recurrence rate cannot compare with Hiatt et al. and Dussol et al. And Borghi et al reported normal-calcium, low protein, low-salt diet had recurrences did reduce the recurrence compared to normal-calcium diet (12/60 vs 20/60; RR = 0.49, 95%CI = 0.24–0.98) [19]. And, only two studies [17, 21] were included to evaluate the effect of only trials of low protein with or without high fiber diet method. A random-effects

**Table 1. Characteristics of included studies.**

| No. | Author | year | Experimental group | | | | | Control group | | | | | Follow up time (year) |
|---|---|---|---|---|---|---|---|---|---|---|---|---|---|
| | | | Number (n) | Age (Mean ±SD) | Male (n) | Type of stone | Method | Number (n) | Age (Mean ±SD) | Male (n) | Stone type | Method | |
| 1 | Dussol | 2008 | 115 | 44±12 | 60 | Calcium oxalate | 55 patients receive low doses protein (<15% of total energy) and 65 patients receive high doses fiber(>25g) per day | 63 | 45±11 | 38 | Calcium oxalate | Normal diet | 4 |
| 2 | Sarica | 2006 | 12 | - | - | Calcium oxalate | Enforced fluid intake (achieve to more than 2.5 liters of urine per day) | 9 | - | - | Calcium oxalate | Nomal diet | 3 |
| 3 | Borghi | 2002 | 60 | 44.8 ±9.2 | 60 | Calcium oxalate/ Calcium phosphate | Normal Ca with reduced protein 52g/d and salt 50 mmol/d | 60 | 45.4±10.9 | 60 | Calcium oxalate | Traditional low Ca diet | 5 |
| 4 | Kocvara | 1999 | 113 | 18–72 | 59 | Calcium stone | Special dietary regimens with low proteins, purine, oxalate and high fiber. | 94 | 18–72 | 37 | Calcium stone | Normal diet | 3 |
| 5 | Hiatt | 1996 | 50 | 43.1 ±1.5 | 36 | Calcium oxalate | 56–64 g/d protein, 75 mg/d purine, fiber supplement and 6–8 glasses of water | 49 | 42.9 ± 1.4 | 42 | Calcium oxalate | Fluid intake | 4.5 |
| 6 | Borghi | 1996 | 99 | 42.2 ±11.6 | 70 | Calcium oxalate | Enforced fluid intake (achieve to more than 2 liters of urine per day) | 100 | 40.4± 13.2 | 60 | Calcium stone | Normal diet | 5 |

model was used for analyzing the primary outcomes. Our result showed that dietary intervention does not decrease the recurrence of stone upon comparing with control groups (RR = 2.32, 95% CI = 0.42–12.85; P = 0.34) with significant heterogeneity among the studies ($I^2$ = 81%, P = 0.02) (Fig 3). However, the fluid intake was found to have a positive effect on the prevention of stone recurrence (RR = 0.39, 95% CI = 0.19–0.80; P = 0.01) with insignificant heterogeneity among the studies ($I^2$ = 9%, P = 0.30, Fig 4).

Three dietary groups reported the rate of withdrawal of patients. Though the result is statistically insignificant (RR = 0.76, 95% CI = 0.59–0.98; P = 0.03) insignificant heterogeneity was observed among the studies ($I^2$ = 0%, P = 0.81, Fig 5). All the studies showed that intervention groups have a higher withdrawal rate. All patients in fluid intervention froup finish the follow up.

## Secondary outcome: Variables of urine composition

The Urinary variables at baseline were reported in four studies [17–20]. The urine volume was reported in all four studies and one fluid intake group were changed significantly. The 24-hour urine sodium, calcium, citrate, urea, and sulfate were reviewed in other dietary therapy groups. Borghi et al. reported urine sodium and oxalate changed in their study as their was a unique one which provided both low protein and low salt [19]. And Kocvara et al. showed that calcium and oxalate changed in their study [20]. Two studies reported the relative oxalate saturation and both their changes are significant [18, 19].

## Sensitivity analysis

Among all outcomes, the dietary intervention showed a significant heterogeneity ($I^2$ = 81%, P = 0.02). The heterogeneity may come from the study design as the specific dose of protein are different.

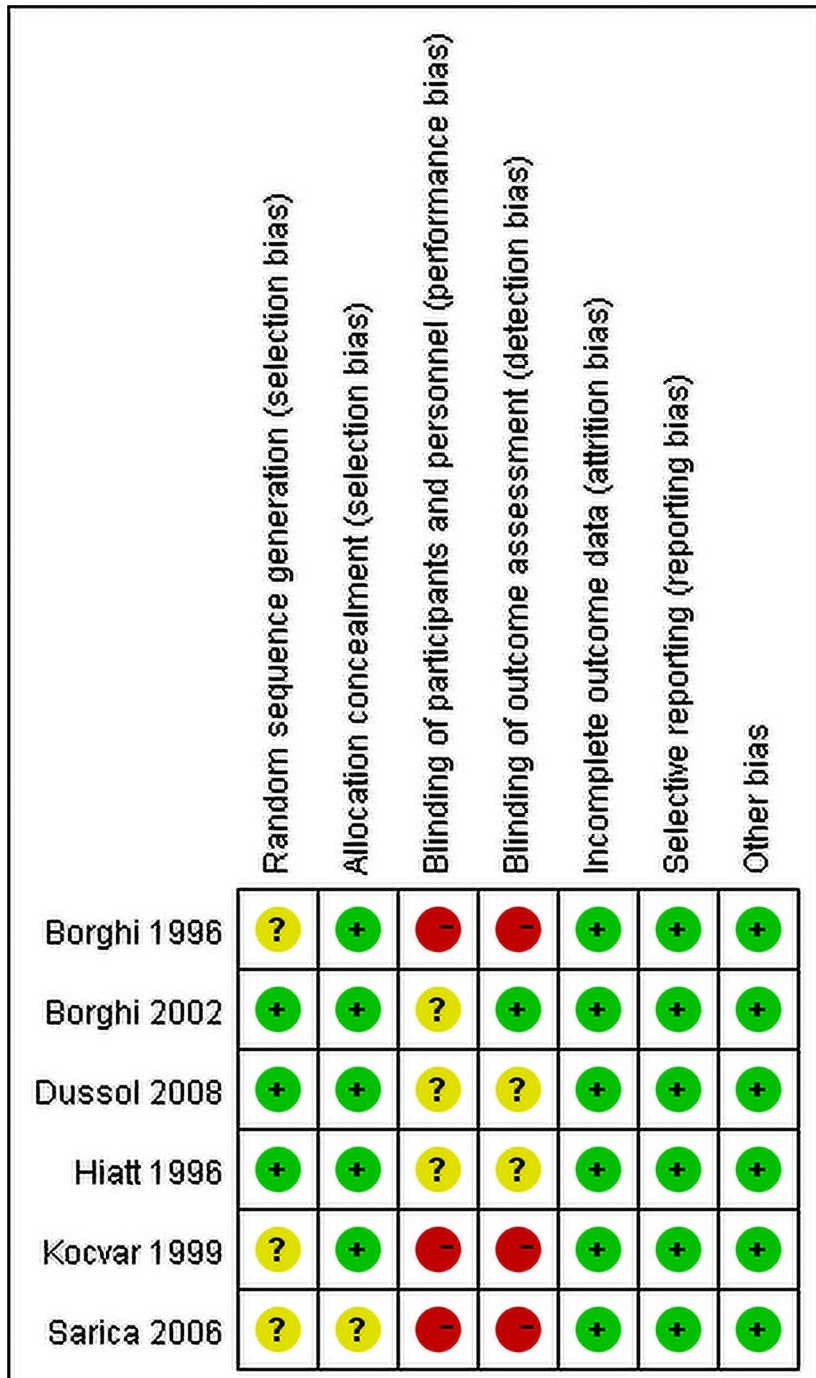

**Fig 2. Risk of bias assessment in included studies.**

## Discussion

Prevention of kidney stone recurrence is very complex and controversial. A cohort study showed that about half of the nephrolithiasis were ascribed to lifestyle factors [11]. Particularly, diet interventions were assumed as an efficient technique to prevent urinary stone formation

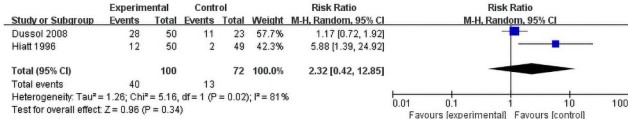

**Fig 3. Forest plot for the meta-analysis of recurrence of only low protein with or without high fiberdiet.**

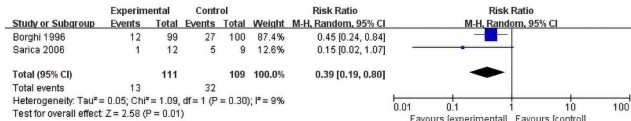

**Fig 4. Forest plot for the meta-analysis of recurrence rate of liquid intake.**

and its recurrence. However, due to the lack of high-quality original research, no conclusive consensus or guideline was drawn for the effect of secondary prevention for stone formation [23]. Here, we conducted this study to review the RCTs reporting the effects of different dietary interventions for the prevention of recurrent stone formation and the changes in urine compositions in patients with urinary stone disease.

The results of this study indicate that water intake can reduce stone recurrence. Meanwhile low protein with or without fiber does not affect recurrence, but low Na, normal Ca diet has a marked reduction effect on recurrence. All included studies use reduced protein intake for patients as an intervention process. Higher animal protein intake may result in decreased citrate and urine PH and increased urinary excretion of calcium and uric acid which may potentially favor the formation of stones [24]. But the present study unexpectedly contradicted this hypothesis. In one subgroup of the study by Dussol *et al.*, the only intervention is the restriction of the consumption of animal protein, but the recurrence rate was 48% (11 of 23) in the intervention group which is almost the same as that for the control group (48%;11 of 23) [17]. Also, Hiatt et al. reported that dietary intervention combining low protein, high fiber, and fluid intake has no advantage compared to fluid intake only [21]. An alternative explanation is that different types of protein may affect the recurrence of kidney stones differently. One large cohort study showed that red and processed meats increase the risk of stone formation than other animal or vegetable protein [25]. This is difficult to control in long-term RCTs. Furthermore, protein intake causes an increase in excretion of lithogenic materials such as calcium oxalate, sulfate, and uric acid, and causes a reduction in the excretion of citrate [26, 27]. But our study shows that these changes are not obvious (Table 2). Not all stone formation compositions of the urine increase correspondingly and urine citrate decreased statistically insignificant. This may suggest that low-protein intake is not low enough to provide the desired effect. Dussol *et al.* assumed that their intervention with low animal protein (17% total energy) is not low enough compared to the people who often follow a Mediterranean diet that is relatively

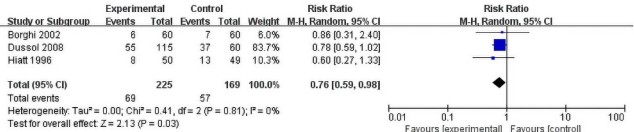

**Fig 5. Forest plot for the meta-analysis of withdraw rate.**

**Table 2. Urinary compositions variables in baseline and the long term follow up.**

| Study | Follow up | Method | Urine volume (L/day) | | Sodium (mmol/day) | | Calcium (mmol/day) | | Oxalate (mmol/day) | | Citrate (mmol/day) | | Urea (mmol/day) | | Sulfate (mmol/day) | | Relative oxalate saturation | |
|---|---|---|---|---|---|---|---|---|---|---|---|---|---|---|---|---|---|---|
| | | | Baseline | Result | Baseline | Result | Baseline | Result | Baseline | Result | Baseline | Result | Baseline | Result | Baseline | Result | Baseline | Result |
| Dussol | 3 year | Low protein | 1.8±0.6 | 1.9±0.8 | 149±44 | 171±71 | 6.8±3.1 | 7.0±3.5 | 0.30±0.1 | 0.29±0.1 | 2.9±1.9 | 2.9±1.5 | 381±95 | 359±135 | 4.3±1.9 | 3.1±2.4 | | |
| | | High fiber | 2.0±0.7 | 1.8±0.6 | 163±58 | 154±55 | 6.9±3.7 | 7.0±3.5 | 0.31±0.2 | 0.32±0.1 | 3.3±3.2 | 2.1±1.2 | 354±93 | 361±117 | 4.6±2.8 | 4.8±3.6 | | |
| Borghi | 3 year | Low protein Low salt | 1.9±6.7 | 2.1±5.2 | 205±64 | 127±66[b] | 7.0±3.4 | 6.6±2.4 | 0.42±0.1 | 0.33±0.1[b] | | | 505±142 | 447±113 | 2.8±0.8 | 2.5±0.6 | 6.7±4.5 | 4.5±2.9[a] |
| Kocvara | 3 year | Special diet | 2.4±6.5 | 2.4±6.5 | | | 5.1±2.4 | 6.4±2.8[b] | 0.35±0.2 | 0.42±0.2[a] | 3.0±1.5 | 3.2±1.9 | | | | | | |
| Borghi | 5 year | Fluid intake | 1.1±0.2 | 2.6±0.7[b] | | | | | | | | | | | | | 1.8±1.7 | 1.2±1.0[b] |

[a]: P<0.05 compared to the baseline;

[b]: P<0.01 compared to the baseline

poor in animal protein [17]. However, Borghi et al. added low Na with normal Ca in diet reduced the probability of stone recurrence which indicated the positive effect of low Na for prevention the stone [19].

The definition of "low" for "low protein" is still unclear. Another reason is that each patient's daily diet structure is complex as a low protein diet may change the urine composition in a very short follow-up period but may not in an interventional long-term trial. Furthermore, specific dietary instructions should be corrected according to repeated metabolic measurements and low protein intake could produce other metabolic disorders [12]. Kocvara et al. found a positive result in their study as each patient underwent a personalized metabolic assessment and specific diet regime. And patients with hypercalciuria and hyperuricemia took a low protein diet can significantly reduce the recurrence of stones [20]. Rotily et al. also found in a 24-hour urine composition analysis that patients with hypercalciuria benefit more from a low protein diet as the urea and calcium outputs were observed only among hypercalciuria patients [28]. Therefore, we can conclude that a low-protein diet does not prevent the recurrence of stones in all patients, but it may benefit patients with hypercalciuria. Thus, metabolic evaluation is strongly recommended for patients before choosing low protein diet therapy. The withdrawal rate in low protein is higher than in the control group. Also, Dussol et al. and Borghi et al. reported that the rate of unwillingness for withdrawal patients to continue the therapy is also higher which indicates poor patient compliance though it is statistically insignificant [17, 19].

Our results show that an increase in dietary fiber intake failed to reduce the recurrence rate of calcium oxalate stones. In the study by Dussol et al., the subgroup taking a high diet only did not succeed in the prevention of recurrence as well as the changes in urine composition [17]. This result is similar to the study by Rotily et al. in which the patients with dietary fiber intake do not reduce the predictive factors of calcium oxalate such as calcium or oxalate outputs [28]. Haitt et al. found that a combination of high fiber and low protein diet did not reduce stone recurrence [21]. Phytate in fibers can reduce hypercalciuria by forming complex molecules with calcium in the gut [29]. Therefore, fiber-rich foods can decrease calcium oxalate supersaturation and increase citrate excretion [30]. Nevertheless, intestinal complexation of calcium might lead to hyperoxaluria [31]. So, the overall effect of high fiber in preventing the recurrence of calcium oxalate stones is unclear. An observational study by Hirvonen et al. showed that a fiber-rich diet may increase the recurrence of kidney stones [32]. Other results of recent cohort studies reported fiber-rich diet might reduced recurrence or no relation for stone formation [33, 34]. The results that fluid intake reduced stone recurrence were not surprising. Increased fluid intake may help prevent the formation of stones by diluting urine components and decreasing urine acidity which is well learned and suggested as the first prevention step for urolithiasis [35]. This concept is consistent with the results of Borghi et al. which showed a significant reduction in the relative supersaturation of calcium oxalate, brushite, and uric acid [18]. It is worth noting that there is no withdrawal patient in two fluid intake groups indicating that water intake is highly associated with patient compliance.

There are a few limitations associated with this study. Firstly, the heterogeneity of dietary prevention is high due to the variation in the intervention standards in the study design. Secondly, the impact of dietary factors on stone recurrence varies from age, gender, race, and region remained unknown due to the lack of related studies. Lastly, the accuracy of our summary may be skewed as there are publication biases due to some unpublished data.

In conclusion, our study shows that the only low protein with or without fiber does not affect recurrence, but low Na, normal Ca diet has a marked effect on reducing recurrence of calcium stone. And fluid intake shows a significant reduction in the recurrence of calcium stone.

## Supporting information

**S1 Checklist. Effect of dietary treatment and fluid intake on the prevention of recurrent calcium stones and changes in urine composition: A meta-analysis and systematic review.** (DOC)

**S1 Table. S1 Table for quality assessment in all the selected studies for systematic review and meta-analysis.** (DOCX)

**S1 Data. All relevant data of this study.** (DOCX)

**S1 Search strategy.** (DOCX)

## Author Contributions

**Conceptualization:** Zhenghao Wang.

**Data curation:** Zhenghao Wang, Yu Zhang.

**Formal analysis:** Zhenghao Wang.

**Methodology:** Zhenghao Wang.

**Project administration:** Zhenghao Wang.

**Resources:** Zhenghao Wang, Yu Zhang.

**Software:** Zhenghao Wang, Yu Zhang.

**Supervision:** Wuran Wei.

**Validation:** Zhenghao Wang.

**Visualization:** Zhenghao Wang.

**Writing – review & editing:** Zhenghao Wang, Yu Zhang.

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
