## [Decision Letter · Decision Letter 0]

26 Jan 2021

PONE-D-20-37694

Effect of dietary treatment and fluid intake on the prevention of recurrent calcium stones and changes in urine composition: a meta-analysis and systematic review

PLOS ONE

Dear Dr. Wei,

Thank you for submitting your manuscript to PLOS ONE. After careful consideration, we feel that it has merit but does not fully meet PLOS ONE’s publication criteria as it currently stands. Therefore, we invite you to submit a revised version of the manuscript that addresses the points raised during the review process.

We look forward to receiving your revised manuscript.

Kind regards,

Tzevat Tefik, MD

Academic Editor

PLOS ONE

Journal Requirements:

2. Please confirm that you have included all items recommended in the PRISMA checklist including the full electronic search strategy used to identify studies with all search terms and limits for at least one database.

"No."

Reviewers' comments:

Reviewer's Responses to Questions

**Comments to the Author**

1. Is the manuscript technically sound, and do the data support the conclusions?

Reviewer #1: Partly

Reviewer #2: Yes

2. Has the statistical analysis been performed appropriately and rigorously? 

Reviewer #1: I Don't Know

Reviewer #2: I Don't Know

3. Have the authors made all data underlying the findings in their manuscript fully available?

Reviewer #1: Yes

Reviewer #2: Yes

4. Is the manuscript presented in an intelligible fashion and written in standard English?

Reviewer #1: Yes

Reviewer #2: Yes

5. Review Comments to the Author

Reviewer #1: Is the manuscript technically sound, and do the data support the conclusions?

With respect to their ability to adhere to guidelines for performance of meta-analyses, I do not have enough experience to judge. However, they have combined studies that do not test the same interventions, which leads to inappropriate conclusions. The Results section dealing with the ability of diet interventions to prevent stone recurrence is poorly written and confusing, and the figure that illustrates their findings lacks one of the 4 studies being analyzed. The data in the underlying papers partially supports their conclusions; for example the 2 studies of increased fluid do show decreased stone recurrence. The studies of diet alone however are not analyzed appropriately, and thus do not support their conclusions.

Is the manuscript presented in an intelligible fashion and written in standard English?

It could use some editorial polishing to improve the grammar. Also a few typos. The primary results section is hard to follow.

The authors have done a meta-analysis of studies in which dietary interventions were done for nephrolithiasis, with recurrent stone formation as the primary outcome. They found 6 randomized controlled trials of diet or fluid intake that satisfied their criteria. The 2 studies of increased fluid intake appear to be comparable, and both show decreased stone recurrence in the group with increased fluid intake. However the authors conclude that diet interventions did not significantly reduce stone recurrence.

Comments for the investigators:

2. 1. The Introduction includes the statement that “medication has a therapeutic effect on the prevention of recurrent stone formation by reducing the tubular reabsorption or increasing the intestinal reabsorption of calcium.” This is incorrect, and in fact these changes in renal or intestinal calcium handling would actually lead to higher urine calcium excretion. The effect of thiazide is to increase renal calcium reabsorption, in addition to a modest effect to decrease gut calcium absorption. The effect of citrate is multifactorial, and includes its ability to chelate calcium in the urine, and to poison calcium crystal surfaces; it may also lower urine calcium by increasing renal calcium reabsorption.

3. 2. The authors used the Jadad scale to assess for study quality. The 3 items in the scale assess randomization (2 possible points), blinding (2 possible points), and accounting for all patients (1 point). The authors need to justify use of this scale for long-term dietary/fluid studies in which blinding is not attempted or likely to succeed. If possible, a more appropriate scale should be used which was designed for this purpose.

4. 3. The diet studies are not strictly comparable with respect to the interventions carried out. Diet alone is not an intervention, rather the specific type of diet interventions is what is being tested. For background: The 4 diet studies comprise 9 study arms. Three studies have 2 arms, while Dussol has three (control and 2 interventions). The control arms in three studies (Hiatt, Kocvara, and Dussol) are somewhat similar (normal calcium intake, encourage fluids) while the interventions vary. Two of these studies specifically study animal protein and fiber in the diet. In Hiatt, the intervention is low animal protein, increased fruits and vegetables, and bran; in Dussol the 2 intervention arms compare low animal protein (arm 1) with increased fruits and vegetables and whole grain (arm 2). Patient numbers are quite small in Dussol due to drop outs. In Hiatt, the control arm does better with respect to stone recurrence, while in Dussol the three arms all have a very high recurrence rate; both studies follow patients for about 4 years. Overall, it seems fair to say that low animal protein diet, with or without high fiber (or high fiber diet alone), does not prove to lower stone recurrence better than a diet with high fluid and normal calcium intake alone, based on these 2 studies. The recurrence rates per year do vary a lot in these studies, and the authors do not attempt to report them. I suggest that the authors use only these 2 studies to evaluate the effect of protein restriction and fiber.

5. 4. The diet studies by Kocvara and by Borghi are not so readily compared with Hiatt and Dussol. While the control arm of Kocvara is similar, what is being tested in the intervention arm is not a specific diet, but rather the concept that diet prescription based on metabolic workup, rather than generic advice, is beneficial, and the study results bear this out. Thus this study does not really test a specific diet per se, but rather whether diets should be tailored or generic. This study does not really belong in this analysis, as the type of dietary advice is unknown in the intervention group, being individualized to each patient.

6. 5. The diet study by Borghi compares low calcium diet (a unique diet arm which differs markedly from the control arms in other studies) to a diet with low sodium intake (not studied in any of the other studies), along with normal calcium intake, and modestly low animal protein intake. The fact that sodium restriction is the main intervention is clearly shown in Table 2, in which urine Na clearly falls on follow-up, while urine urea does not. The authors should not analyze this study as a low protein trial, because it was sodium restriction which was the major intervention. At 5 years (the study endpoint) the recurrence rates are much lower in the low sodium arm compared to low calcium diet This study is not a simple low protein trial, and should not be included with the 2 explicitly low protein diet trials. In addition, the patient group in this study differs from the other trials, in that idiopathic hypercalciuria, rather than only calcium stone formation, was an explicit study inclusion criteria. The authors should rectify the errors in analysis of this trial, and analyze it as a unique study

7. 6. The paragraph titled Primary Outcome in results describes an analysis which seems to include all the diet studies (ref 17, 20-22) but Figure 2 shows only Dussol, Hiatt, and Kocvar.

Overall, the analysis does not compare like to like, lumping non-comparable studies together, Its handling of the Borghi diet trial is particularly unclear, and therefor the conclusion that dietary intervention does not decrease recurrence of calcium stone does not appear to be valid. A more limited conclusion, that low protein diet, with or without increased fiber, has not proven to lower calcium stone recurrence, may be justified.

Reviewer #2: I believe that this paper is of value because it confirms on the one hand that fluid intake affects stone formation and cast into doubt that urinary risk factors are important in this regard. I suggest that the authors emphasize a few findings in their discussion. Firstly, authors should emphasize that their study investigates as a primary outcome stone RECURRENCE per se as opposed to stone RISK FACTORS. This is important. Second, the findings which refute popular views are equally important, namely that diet does not affect recurrence, that diet does not affect urine composition and that the latter does not affect the former. These are new and provocative findings and will initiate a re-think among stone researchers. Indeed, authors can challenge readers by posing a question about the clinical value of dietary interventions in the first place. Yes, this will be controversial, but that's what differentiates this study from others.

6. PLOS authors have the option to publish the peer review history of their article (what does this mean?). If published, this will include your full peer review and any attached files.

Reviewer #1: No

Reviewer #2: **Yes: **Allen Rodgers

---

## [Author Response · Author response to Decision Letter 0]

10 Feb 2021

Reviewer #1：

1. The Introduction includes the statement that “medication has a therapeutic effect on the prevention of recurrent stone formation by reducing the tubular reabsorption or increasing the intestinal reabsorption of calcium.” This is incorrect, and in fact these changes in renal or intestinal calcium handling would actually lead to higher urine calcium excretion. The effect of thiazide is to increase renal calcium reabsorption, in addition to a modest effect to decrease gut calcium absorption. The effect of citrate is multifactorial and includes its ability to chelate calcium in the urine, and to poison calcium crystal surfaces; it may also lower urine calcium by increasing renal calcium reabsorption.

Response: We are very sorry for the mistake and appreciated for the reviewer’s correction. We have corrected this part as the comment (Currently, medication has a therapeutic effect on the prevention of recurrent stone formation by increasing renal calcium reabsorption, decreasing gut calcium absorption, chelating calcium in the urine or poisoning calcium crystal surfaces.).

2. The authors used the Jadad scale to assess for study quality. The 3 items in the scale assess randomization (2 possible points), blinding (2 possible points), and accounting for all patients (1 point). The authors need to justify use of this scale for long-term dietary/fluid studies in which blinding is not attempted or likely to succeed. If possible, a more appropriate scale should be used which was designed for this purpose.

Response: Thanks for this professional comments. We have changed the assessment tool. We used the risk bias tool of Cochrane Handbook for Systematic Reviews of Intervention which is more evidence based and we changed this part correspondingly.

3. The diet studies are not strictly comparable with respect to the interventions carried out. Diet alone is not an intervention, rather the specific type of diet interventions is what is being tested. For background: The 4 diet studies comprise 9 study arms. Three studies have 2 arms, while Dussol has three (control and 2 interventions). The control arms in three studies (Hiatt, Kocvara, and Dussol) are somewhat similar (normal calcium intake, encourage fluids) while the interventions vary. Two of these studies specifically study animal protein and fiber in the diet. In Hiatt, the intervention is low animal protein, increased fruits and vegetables, and bran; in Dussol the 2 intervention arms compare low animal protein (arm 1) with increased fruits and vegetables and whole grain (arm 2). Patient numbers are quite small in Dussol due to drop outs. In Hiatt, the control arm does better with respect to stone recurrence, while in Dussol the three arms all have a very high recurrence rate; both studies follow patients for about 4 years. Overall, it seems fair to say that low animal protein diet, with or without high fiber (or high fiber diet alone), does not prove to lower stone recurrence better than a diet with high fluid and normal calcium intake alone, based on these 2 studies. The recurrence rates per year do vary a lot in these studies, and the authors do not attempt to report them. I suggest that the authors use only these 2 studies to evaluate the effect of protein restriction and fiber. The diet studies by Kocvara and by Borghi are not so readily compared with Hiatt and Dussol. While the control arm of Kocvara is similar, what is being tested in the intervention arm is not a specific diet, but rather the concept that diet prescription based on metabolic workup, rather than generic advice, is beneficial, and the study results bear this out. Thus this study does not really test a specific diet per se, but rather whether diets should be tailored or generic. This study does not really belong in this analysis, as the type of dietary advice is unknown in the intervention group, being individualized to each patient.

Response：Thanks for this comment. We have removed the Kocvara et al. for the analysis for recurrent rate and only take Hiatt and Dussol into primary outcome as comment. And the result did not changed as the new result showed that dietary intervention does not decrease the recurrence of stone upon comparing with control groups (RR = 2.32, 95% CI = 0.42–12.85; P = 0.34) with significant heterogeneity among the studies (I2 = 81%, P = 0.02). Meanwhile, we keep Kocvara et al in our study because the data of withdraw rate and changes in the urine compositions. We have corrected this part in our study accordingly.

4. The diet study by Borghi compares low calcium diet (a unique diet arm which differs markedly from the control arms in other studies) to a diet with low sodium intake (not studied in any of the other studies), along with normal calcium intake, and modestly low animal protein intake. The fact that sodium restriction is the main intervention is clearly shown in Table 2, in which urine Na clearly falls on follow-up, while urine urea does not. The authors should not analyze this study as a low protein trial, because it was sodium restriction which was the major intervention. At 5 years (the study endpoint) the recurrence rates are much lower in the low sodium arm compared to low calcium diet This study is not a simple low protein trial, and should not be included with the 2 explicitly low protein diet trials. In addition, the patient group in this study differs from the other trials, in that idiopathic hypercalciuria, rather than only calcium stone formation, was an explicit study inclusion criteria. The authors should rectify the errors in analysis of this trial, and analyze it as a unique study

Response: This is a very good comment. Sorry for ignorance of the effect of a low-salt diet on changes in the composition of urine in Borghi et al. study. And we have added the low salt diet in table two. And we have corrected the result part and described uniqueness of this study.

5. The paragraph titled Primary Outcome in results describes an analysis which seems to include all the diet studies (ref 17, 20-22) but Figure 2 shows only Dussol, Hiatt, and Kocvar.

Response: Sorry for the mistake we made. Though there are four studies reported the diet method, only three study reported the recurrence rate. Actually, Borghi et al. just reported the changes of urinary composition thus only three studies were included into the analysis (ref 16,19,20; Figure 2) and we have corrected this part.

6.Overall, the analysis does not compare like to like, lumping non-comparable studies together, Its handling of the Borghi diet trial is particularly unclear, and therefore the conclusion that dietary intervention does not decrease recurrence of calcium stone does not appear to be valid. A more limited conclusion, that low protein diet, with or without increased fiber, has not proven to lower calcium stone recurrence, may be justified.

Response: Thanks for this important comment. After re-examining our conclusions, we realized that they were too general to be drawn. Therefore, the limited conclusion is needed and we have correct according to this comments.

Reviewer #2: 

1.Firstly, authors should emphasize that their study investigates as a primary outcome stone RECURRENCE per se as opposed to stone RISK FACTORS. This is important.

Response: It is a very good comment which may make our study clearer. After careful re-read for our study, we changed the description like “increase the risk of stone” into “increase the recurrence of stone”; “reduced risk or no relation for stone formation” into “reduced recurrence or no relation for stone formation” to emphasize the primary outcome stone recurrence according to comment. Meanwhile, our introduction and conclusion part also emphasize the aim of this study is to “investigating the effects of dietary treatment and fluid intake on the prevention of recurrent calcium stones”.

2.Second, the findings which refute popular views are equally important, namely that diet does not affect recurrence, that diet does not affect urine composition and that the latter does not affect the former. These are new and provocative findings and will initiate a re-think among stone researchers. Indeed, authors can challenge readers by posing a question about the clinical value of dietary interventions in the first place. Yes, this will be controversial, but that's what differentiates this study from others.

Response: Thank you for your appreciation and this very good comment on revision. We reorganized part of introduction and add this sentence “Previous studies have shown that formation of a renal stone is closely related to dietary regimes [12, 13]. Nevertheless, an evidence-based study of clinical value in dietary therapy in urinary stone recurrence is still lacked. Furthermore, new studies with more detailed data at high evidence level are reported.” With this sentence, we pose a question about the clinical value of dietary interventions in the first place and differentiates this study from others as the comments.

Acknowledgment: Our deepest gratitude goes to the reviewers and editors for their careful work and thoughtful suggestions that have helped improve this paper substantially. We also thanks for both their meaningful comments and recognition to our study again.

---

## [Decision Letter · Decision Letter 1]

9 Mar 2021

PONE-D-20-37694R1

Effect of dietary treatment and fluid intake on the prevention of recurrent calcium stones and changes in urine composition: a meta-analysis and systematic review

PLOS ONE

Dear Dr. Wei,

Thank you for submitting your manuscript to PLOS ONE. After careful consideration, we feel that it has merit but does not fully meet PLOS ONE’s publication criteria as it currently stands. Therefore, we invite you to submit a revised version of the manuscript that addresses the points raised during the review process.

We look forward to receiving your revised manuscript.

Kind regards,

Tzevat Tefik, MD

Academic Editor

PLOS ONE

Reviewers' comments:

Reviewer's Responses to Questions

**Comments to the Author**

1. If the authors have adequately addressed your comments raised in a previous round of review and you feel that this manuscript is now acceptable for publication, you may indicate that here to bypass the “Comments to the Author” section, enter your conflict of interest statement in the “Confidential to Editor” section, and submit your "Accept" recommendation.

Reviewer #1: (No Response)

Reviewer #2: All comments have been addressed

2. Is the manuscript technically sound, and do the data support the conclusions?

Reviewer #1: No

Reviewer #2: Yes

3. Has the statistical analysis been performed appropriately and rigorously? 

Reviewer #1: I Don't Know

Reviewer #2: I Don't Know

4. Have the authors made all data underlying the findings in their manuscript fully available?

Reviewer #1: Yes

Reviewer #2: Yes

5. Is the manuscript presented in an intelligible fashion and written in standard English?

Reviewer #1: Yes

Reviewer #2: Yes

6. Review Comments to the Author

Reviewer #1: The authors have responded to the prior comments, but I have some comments about the current manuscript.

1. In the Introduction, the authors state that “..an evidence-based study of clinical value in dietary therapy in urinary stone recurrence is still lacked”. It is unclear why the authors make this statement, as they are reviewing a number of evidence based studies. What do they feel is lacking?

2. In the section on selection criteria, they specify that the in the studies they selected “patient had stone that has been cleared by surgery”. This is incorrect. The studies they are using do not require surgical removal of stones, only stone passage (or removal) or in some cases visualization on Xray.

3. In the section on Primary outcome, they state that 5 of the studies reported stone recurrence, omitting the Borghi trial of low sodium diet. This is incorrect. That study does report stone recurrence: “Twenty-three of the 60 men on the low-calcium diet and 12 of the 60 on the normal-calcium, low protein,

low-salt diet had recurrences. The cumulative incidence of recurrent stones in the two groups is

shown in Figure 2. The relative risk of a recurrence among the men in the normal-calcium, low-protein,

low-salt group, as compared with the men in the low calcium group, was 0.49 (95 percent confidence interval,

0.24 to 0.98; P=0.04).” This data should be added to the section on primary outcome.

4. The authors continue to refer to “diet method” as the intervention being studied. This is incorrect. The authors need to specify what type of diet is being evaluated. In the case of Hiatt and Dussol, it is low protein with or without high fiber. They should modify the statement in “Primary outcome:recurrence rate” to make clear that this type of diet did not reduce stone recurrence rate.. They should also make clear that the Borghi study of normal calcium, low sodium diet did reduce recurrence rate. They should also modify Figure 3 accordingly, although given that there is only one study of low Na, normal Ca diet it is not clear that Borghi can be added to the plot. However, the legend should make clear that only trials of low protein with or without high fiber are included.

5. The materials do not include Figure legends.

6. The authors state in the Discussion that changes in diet do not affect recurrence. This is incorrect. Low protein diet does not affect recurrence, while low Na, normal Ca diet has a marked effect on recurrence. This statement must be changed.

Reviewer #2: The main interest in this study is that its findings are contrary to popular dogma. As such it is likely to attract a lot of attention

7. PLOS authors have the option to publish the peer review history of their article (what does this mean?). If published, this will include your full peer review and any attached files.

Reviewer #1: No

Reviewer #2: **Yes: **Allen Rodgers

---

## [Author Response · Author response to Decision Letter 1]

10 Mar 2021

Reviewer #1: 

1. In the Introduction, the authors state that “..an evidence-based study of clinical value in dietary therapy in urinary stone recurrence is still lacked”. It is unclear why the authors make this statement, as they are reviewing a number of evidence-based studies. What do they feel is lacking?

Response：This is a very good comment. What we mean is higher evidence-based study like systematic review and meta-analysis and we have changed this part. We have changed the vague expression and thank you for your correction.

2. In the section on selection criteria, they specify that the in the studies they selected “patient had stone that has been cleared by surgery”. This is incorrect. The studies they are using do not require surgical removal of stones, only stone passage (or removal) or in some cases visualization on Xray.

Response：Thanks for comment. We have checked our origin study design and found the mistake in our description. The inclusion criteria is patient had a stone history and was diagnosed by surgical removal, stone passage, or by imaging systems. And we correct this part in our study.

3. In the section on Primary outcome, they state that 5 of the studies reported stone recurrence, omitting the Borghi trial of low sodium diet. This is incorrect. That study does report stone recurrence: “Twenty-three of the 60 men on the low-calcium diet and 12 of the 60 on the normal-calcium, low protein, low-salt diet had recurrences. The cumulative incidence of recurrent stones in the two groups is shown in Figure 2. The relative risk of a recurrence among the men in the normal-calcium, low-protein, low-salt group, as compared with the men in the low calcium group, was 0.49 (95 percent confidence interval, 0.24 to 0.98; P=0.04).” This data should be added to the section on primary outcome.

Response：Thanks for this very good comment. We ignore the result of Borghi trial and made an incomplete conclusion. And we have added the Borghi trail result into the primary outcome. And correcting the our related conclusion for diet intervention into two part：1. Low protein with or without high fiber do not affect the recurrence； 2. normal calcium, low sodium diet did reduce recurrence rate, as comment. Therefore, our conclusions are completer and more accurate.

4. The authors continue to refer to “diet method” as the intervention being studied. This is incorrect. The authors need to specify what type of diet is being evaluated. In the case of Hiatt and Dussol, it is low protein with or without high fiber. They should modify the statement in “Primary outcome: recurrence rate” to make clear that this type of diet did not reduce stone recurrence rate. They should also make clear that the Borghi study of normal calcium, low sodium diet did reduce recurrence rate. They should also modify Figure 3 accordingly, although given that there is only one study of low Na, normal Ca diet it is not clear that Borghi can be added to the plot. However, the legend should make clear that only trials of low protein with or without high fiber are included.

Response：Thanks for this comment. Hiatt and Dussol study show low protein with or without high fiber do not affect the recurrence. And Borghi study shows normal calcium, low sodium diet did reduce recurrence rate. We changed the primary result, figure legend and conclusion accordingly.

5. The authors state in the Discussion that changes in diet do not affect recurrence. This is incorrect. Low protein diet does not affect recurrence, while low Na, normal Ca diet has a marked effect on recurrence. This statement must be changed.

Response：Thanks for this detailed comment. This statement is too general which is incorrect, and we have corrected the statement correspondingly as comment in discussion part (Second paragraph; line 2) and conclusion part. 

Acknowledgment: Thanks again. Our deepest gratitude goes to the you for your careful work and detailed comment that have helped improve the quality of our study.

Reviewer #2: The main interest in this study is that its findings are contrary to popular dogma. As such it is likely to attract a lot of attention.

Acknowledgment: Thank you for your recognition of our article. At the same time, we also deeply thank you for your careful and detailed review work. It is your very detailed and professional comments that make our articles more scientific and more complete.

---

## [Editor Report · Decision Letter 2]

5 Apr 2021

Effect of dietary treatment and fluid intake on the prevention of recurrent calcium stones and changes in urine composition: a meta-analysis and systematic review

PONE-D-20-37694R2

Dear Dr. Wei,

We’re pleased to inform you that your manuscript has been judged scientifically suitable for publication and will be formally accepted for publication once it meets all outstanding technical requirements.

Kind regards,

Tzevat Tefik, MD

Academic Editor

PLOS ONE
---

## [Editor Report · Acceptance letter]

8 Apr 2021

PONE-D-20-37694R2 

Effect of dietary treatment and fluid intake on the prevention of recurrent calcium stones and changes in urine composition: a meta-analysis and systematic review 

Dear Dr. Wei:

I'm pleased to inform you that your manuscript has been deemed suitable for publication in PLOS ONE. Congratulations! Your manuscript is now with our production department. 

Kind regards, 

on behalf of

Dr. Tzevat Tefik 

Academic Editor

PLOS ONE